# A Narrative Review of Recent Finite Element Studies Reporting References for Elastic Properties of Zirconia Dental Ceramics

**Layla A. Abu-Naba'a**

Department of Prosthodontics, Faculty of Dentistry, Jordan University of Science and Technology, Irbid 21110, Jordan; laabunabaa@just.edu.jo

**Abstract:** For fixed dentures, new generations of zirconia with diverse characteristics and design choices are of significant interest. Although in vitro studies and finite element analysis (FEA) studies have been published, comprehension of various new methods of material testing and analysis remains insufficient. Young's modulus and Poisson's ratio of the desired materials must be available for investigators to validate FEA investigations that are accompanied by mechanical testing. The aim of this narrative review was to find recent FEA studies that report these values for newly developed commercial CAD/CAM zirconia restorative materials and compile them in a data list. A PubMed search was performed (English articles; 2018–2023; keywords: FEA, finite element, zirconia). Full-text articles (157) were examined, including studies (36) reporting the commercial materials' names, Young's modulus, and Poisson's ratio. Only 21 studies had the source of their values referenced. A list of the materials and values used in these studies was compiled. Researchers are highly recommended to trace back references to determine the origins of these values for commercial materials. New research is encouraged to test the ever expanding list of new commercial esthetic monolithic CAD/CAM zirconia materials, as well as their different translucencies, to report their Young's modulus and Poisson's ratio.

**Keywords:** monolithic ceramic; translucent zirconia; multi-layered zirconia; CAD/CAM; Young's modulus; Poisson's ratio; finite element analysis

## 1. Introduction

Prosthetic materials, such as implants, abutments, and dental restorations, must serve for a long period in the oral environment. Long-term, controlled, randomized, and double-blind clinical trials are agreed to be the gold standard to confirm this [1]. In vitro studies simulate isolated or combined clinical conditions allowing for the prediction of material behavior under the effects of each clinical variable without the involvement of patients. Such studies require significantly less time and funds than clinical studies [2]. The use of digital model simulations and non-invasive investigations is another way to validate the deployment of newly manufactured lines of previously authorized materials that have not yet been completely clinically evaluated. These virtual testing approaches minimize the time and expense requirements even further in comparison with in vitro testing [2]. In order to design such studies, numerical values for the various materials' properties must be available for equations and computer-driven simulations for the object being modeled [3]. Several simulation methods rely on photoelastic strain gauge techniques.

One well known example is dental implant research, where finite element analysis (FEA) studies aided in the investigation of stresses exerted on the peri-implant region and in the components of implant-supported restorations. Vertical and transverse stress from mastication yield lateral loading and bending movements and lead to stress variations along the implant body together with surrounding bone. Stress can be static or dynamic. Resultant strain could occur when forces exceed the strength of any component of these restoration–implant–tissue interfaces. This strain could be simulated in virtual models

formulated by computer programs and translated to visual images seen at each part of the model [3,4]. One of these simulation techniques is FEA, which allows for the mathematical conversions and study of mechanical characteristics of a geometrical object and has several applications in dentistry and oral health sciences. It uses a virtual 3D model built by dividing solid objects into several layered elements with geometric shapes similar to clinical situations. This computer-based analysis is then based on the notion of partitioning the structure into a finite quantity of tiny elements that are linked with each other at the corners (i.e., nodes). Each element's mechanical performance may be described in terms of node displacements caused by simulated forces of shear, tension, and elastic deformation using actual data and complicated sequences of vector equations. It integrates the direction and distortion responses to provide an overall response to stress. Results can be in the form of tabulated data, line graphs, charts, and multicolored contour plots. Results also appear as a photoelastic image, where it is possible to evaluate and visualize structural stresses and strain caused by external loading [5].

The advantages of such studies make it popular as FEA is a non-invasive technique that reduces, but does not eliminate, the need for laboratory testing. Even with complicated structures, it takes less time. Results can be understood both physically and mathematically. Elements' parameters can be altered to suit homogeneous or non-homogeneous structures as well. The characteristics of each constituent can even differ depending on the polynomial applied to it. Preoperative, intraoperative, and postoperative simulations are possible and provide more accurate results. Reproducibility has no effect on material qualities. The experiment can be performed as many times as the researcher desires. The finite element procedure's systematic universality makes it a strong and adaptable instrument for a multitude of problems [3–5].

Because the modelization procedure is the most important step in this test, the physical properties of the actual proposed material should correspond to the dimension and material properties installed in the simulation equations [5]. Many studies advanced their modelizing procedures and attempted to provide lists of material values to be used for later studies [6,7]. Non-living mechanical structures such as implants, abutments, and restorations can be digitally modeled to have isotropic, transversely isotropic, orthotropic, and/or anisotropic properties based on the research questions [8]. The relevant material properties in an isotropic material are the same in all directions, resulting in only two independent material constants, such as Young's modulus and Poisson's ratio. Poisson's ratio measures the material's deformation in a direction perpendicular to the direction of applied force, so it is the ratio of the change in width (per unit width) to its length (per unit length) because of the applied strain. Young's modulus, also known as elastic modulus, is the stress–strain ratio and represents the slope in the stress–strain graph for the material [8]. An anisotropic material has material properties that differ depending on the direction. Still, the materials are typically modeled for FEA as homogeneous, isotropic, and linearly elastic [8]. Sometimes, a compromise is necessary because in some studies restorations or implants are modeled without details such as threads, finish lines, cement layers, detailed grooves, boundary conditions, or specific surface morphologies. This will make the computer-designed model simpler, thus reducing complication of the computer-driven calculations. This reduces the cost, time, and technical specification requirements of the computer performing the calculations. However, these simplifications do not reflect the morphology of the restoration–implant components, nor their behavior observed clinically within their surrounding tissue [9].

This leads to the most common drawback for FEA, from a clinical perspective: many features that could directly affect the model accuracy are neglected or ignored by multiple simplifications and assumptions. Furthermore, different study designs and proposed materials add to the diversity of the models and thus the diversity of the results. This prevents us from directly comparing outcomes across these various study designs [10]. Some may argue that the accuracy of the analysis from the perspective of stress distribution is unaffected as long as the models are compared in the same study [4,6,11]. Validation by

conducting confirmative studies with similar results to these FEA simulations determines the model's correctness [9]. These can be actual ex vivo mechanical tests of matching samples using the same parameters and stresses used in the FEA-modelized sample [9].

When planning an FEA study with concurrent mechanical test confirmations, researchers face challenges, particularly when selecting the material for the mechanical test groups from a large array of commercial materials proposed for that clinical indication [9]. Even if all materials are assumed to be homogeneous, the second challenge is determining the elastic properties specific to the commercial materials chosen. These values (Young's modulus and Poisson's ratio) are not provided in all companies' brochures, nor are they easily found in previous study references. The majority of FEA study references use generic values that are repeated across multiple studies [9]. These generic values do not reflect the presence of new materials, translucencies, or layering procedures (all affecting mechanical test results). Using dental crowns as an example, many FEA studies use the term "zirconia" to describe the crown's material and suggest generic values for both Young's modulus and Poisson's ratio. However, the actual clinical choice of materials for such a clinical scenario allows clinicians to select from tens of zirconia-based, modified, or related ceramic restorative materials. Some of these materials are described as ceramic and CAD/CAM but do not contain zirconia [12]. In the last five years, these material options have greatly expanded, with a wide range of materials available for either lab-based or computer-aided design and manufacturing (CAD/CAM) in cosmetic office restorations. Furthermore, the generic name of the material, modification, chemical composition, translucency, and element of stabilization/number (e.g., for yttria) are insufficient to specify the commercial material's actual values [13–16]. So, if the generic numbers for the Young's modulus and Poisson's ratio are used in the FEA simulations, some may argue that drawing conclusions or comparisons between the FEA results and their confirmation study of matching samples is unrealistic [13,15].

The aim of this narrative review is to search for recent FEA studies that report Young's modulus and Poisson's ratio for newly developed commercial CAD/CAM zirconia restorative materials and then to compile and report these values. This allows us to conclude which studies could be recommended as references for investigators designing FEA studies that are paired with mechanical confirmation tests using these zirconia materials.

## 2. Materials and Methods

In February 2023, one researcher conducted an electronic search of PubMed. The terms used to allow the extraction of all relevant studies were finite element, finite element analysis or FEA, and zirconia, and English papers published in 2018 or later were selected. These keywords were chosen because they are the most generic terms and are common across all research related to this topic and because they would generate the largest number of search results.

The search protocol had these further protocols: [((("zirconia s" [All Fields] OR "zirconias" [All Fields] OR "zirconium oxide" [Supplementary Concept] OR "zirconium oxide" [All Fields] OR "zirconia" [All Fields]) AND (((("finite" [All Fields] OR "finitely" [All 'Fields] OR "finiteness" [All Fields]) AND ("element s" [All Fields] OR "elements" [MeSH Terms] OR "elements" [All Fields] OR "element" [All Fields])) OR ("finite element analysis" [MeSH Terms] OR ("finite" [All Fields] AND "element" [All Fields] AND "analysis" [All Fields]) OR "finite element analysis" [All Fields]) OR "FEA" [All Fields])) AND ((english [Filter]) AND (2018:2023 [pdat]))]. Translations that followed were:

zirconia: "zirconia's" [All Fields] OR "zirconias" [All Fields] OR "zirconium oxide" [Supplementary Concept] OR "zirconium oxide" [All Fields] OR "zirconia" [All Fields]

finite: "finite" [All Fields] OR "finitely" [All Fields] OR "finiteness" [All Fields]

element: "element's" [All Fields] OR "elements" [MeSH Terms] OR "elements" [All Fields] OR "element" [All Fields]

finite element analysis: "finite element analysis" [MeSH Terms] OR ("finite" [All Fields] AND "element" [All Fields] AND "analysis" [All Fields]) OR "finite element analysis" [All Fields].

After the papers were identified and full texts were retrieved for appraisal, a paper was further excluded according to the following criteria:

1. It was a review paper, a letter to an editor, a short communication, or not experimental.
2. Zirconia was not included in the test model, but mentioned in the introduction, discussion, or references sections and thus retrieved in the electronic search.
3. FEA was not included in the tests, but mentioned in the introduction, discussion, or references sections and thus retrieved in the electronic search.
4. Zirconia material was assigned or combined to another material but was used for any part of the body other than the in-arch prosthesis or tooth restoration.
5. Zirconia was only an implant or abutment material, and no zirconia restoration was included.
6. If the material describes a category of zirconia without mentioning a specific commercial product's name or manufacturer.
7. The commercial zirconia materials (dental restoration, prosthesis, even frameworks or copings) were not intended to be shaped by CAD/CAM.
8. The full commercial name of the zirconia material was not specified (i.e., the commercial name, manufacturer, city and country).
9. The company's name is included but the specific commercial product's name is not.
10. The commercial name and manufacturer are clear but the values of Young's modulus and Poisson's ratio are not mentioned for that product in the text nor tables.
11. The product is no longer marketed by the manufacturer after checking the manufacturer's website.
12. The zirconia material is only a filler in resin composite (RC) (CAD/CAM) blocks.
13. The study reports the material to be a "zirconia" material but inspecting the chemical composition shows that it is not. If the chemical composition was not present in the paper, the manufacturer's website was used to verify its presence.

## 3. Results

The following numbers were found using the search terms: FEA, zirconia = 124 studies; finite element, zirconia = 185 studies; finite element analysis, zirconia = 343 studies; total = 652). Following the use of the Boolean operators OR, AND, or NOT and removal of any paper prior to 2018, a total of 181 paper titles were identified and 157 full texts were retrieved for appraisal. The text of the articles was analyzed and 120 of them were rejected because they met the exclusion criteria and did not meet the objectives of the current review. Finally, 36 articles were assembled and thoroughly examined for this study (Figure 1) [12,17–52]. Essential information extracted from the texts of the selected articles is listed in Table 1.

The commercial name of a material is important for the researchers performing FEA studies. Many studies stated the term "zirconia" but did not specify the product's name. Some used general descriptive terms such as "a high translucency zirconia", "reinforced lithium silicate", "hybrid ceramic", "multilayered zirconia", "resin ceramic", or "3, 4 or 5 mol% yttria stabilized zirconia polycrystal". Those were excluded as under each heading comes a long list of commercial products under this general terminology. Zirconia material was mentioned in some studies, although it was only a filler in resin composite block or even a trace in the materials' mixture (<1%). Some articles reported the commercial product's name and manufacturer correctly but the values of Young's modulus and Poisson's ratio were not mentioned, and these articles were thus excluded. On the other hand, commercial names were used in some studies but their Young's modulus and Poisson's ratio were reported for the general zirconia values present in many earlier studies. These were not excluded, but discretion should be practiced when using those references.

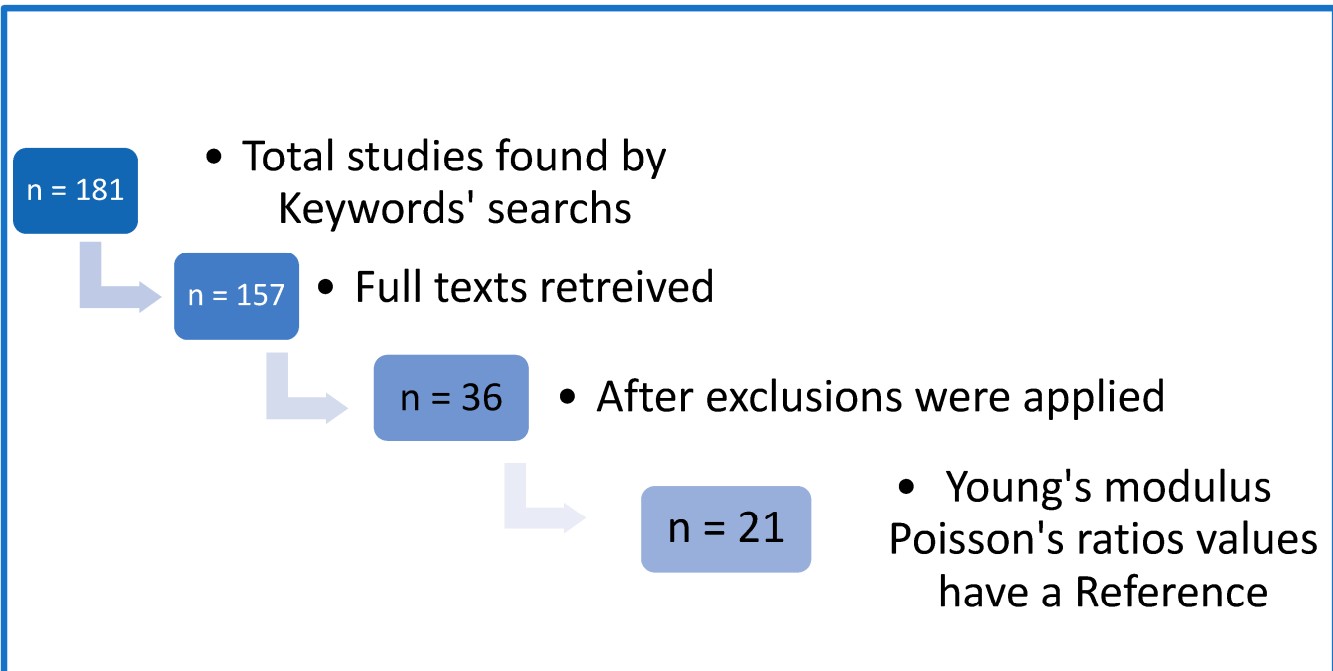

**Figure 1.** A flow chart for the review's methodology.

**Table 1.** After exclusions, the initial screening of FEA studies determines the frequency of essential information being reported or not.

| Information Extracted from the Full-Text Documents | Yes * | No * |
|---|---|---|
| 1. If the article was published and has open access through PubMed. | 11 | 25 |
| 2. If the abstract mentioned the terms "FEA" or "finite element analysis". | 32 | 4 |
| 3. If the commercial name of the material is mentioned in the abstract. | 11 | 25 |
| 4. If the term "monolithic" appears in the abstract. | 15 | 21 |
| 5. The type of restoration or prosthesis investigated is mentioned. | 36 | 0 |
| 6. The chemical composition is mentioned in the text. | 13 | 23 |
| 7. Includes the specific block or disc's shade or subtype. | 12 | 24 |
| 8. The Young's modulus present. | 36 | 0 |
| 9. The value of Poisson's ratio is present. | 35 | 1 |
| 10. The sources of these values' have references. | 21 | 15 |

* Total: 36 full-text articles.

Table 2 contains the compiled list of the articles that report references for their Young's modulus and Poisson's ratio for newly developed commercial CAD/CAM zirconia restoration materials.

**Table 2.** Materials used in FEA studies that have their Young's modulus and Poisson's ratio reported in the articles that reached the final round of inclusion.

| The Reference | Young's Modulus and Poisson's Ratio are Reported for the Following Materials in the Text | Young's Modulus [GPa] | Poisson's Ratio |
|---|---|---|---|
| Abu-Izze, et al. (2018) Fatigue behavior of ultrafine tabletop ceramic restorations [17]. | Lithium silicate zirconia-reinforced ZLS: Vita Suprinity | 65.6 | 0.23 |
| | Hybrid ceramic PIC: Vita Enamic | 34.7 | 0.28 |
| | Panavia F | 3 | 0.35 |
| | Dentin-like material: Epoxy resin G10 | 14.9 | 0.31 |
| | Enamel | 84.1 | 0.33 |
| | Dentin | 18.6 | 0.32 |
| | Structural steel | 200 | 0.30 |
| | Acrylic resin | 26 | 0.38 |
| Archangelo, K.C., et al. (2019) Fatigue failure load and finite element analysis of multilayer ceramic restorations [18]. | Vita In-Ceram YZ | 209.3 | 0.32 |
| | IPS e.max CAD (HT) | 102.7 | 0.21 |
| | Vitablocs Mark II (2 M2C I-40/19) | 64 | 0.25 |
| | Multilink N | 18.6 | 0.28 |
| Bahadirli, G., et al. (2018) Influences of Implant and Framework Materials on Stress Distribution: A Three-Dimensional Finite Element Analysis Study [19]. | Cortical bone | 13.7 | 0.30 |
| | Trabecular bone | 1.37 | 0.30 |
| | Cp grade IV Ti implant | 110 | 0.35 |
| | ZrO, implant and abutment | 220 | 0.30 |
| | TiZr implant | 98 | 0.25 |
| | IPS e.max ZirCAD framework | 220 | 0.30 |
| | IPS e.max CAD framework | 95 | 0.23 |
| | IPS e.max Ceram Feldspathic ceramic | 68 | 0.24 |
| Chirca, O et al. (2021) Adhesive-Ceramic Interface Behavior in Dental Restorations. FEM Study and SEM Investigation. Materials [20]. | Universal RelyX Ultimate clicker | 7.7 | 0.24 |
| | Self-adhesive Maxcem | 4.4 | 0.24 |
| | Double cure Variolink | 8.1 | 0.25 |
| | Pressed IPS e.max press | 82.3 | 0.22 |
| | Carbonat IPS e.max CAD-on | 82.3 | 0.23 |
| | Zirconia Novodent GS | 88 | 0.34 |
| | Dentine | 17 | 0.30 |
| | Enamel | 74 | 0.23 |
| Dal Piva, A.M.O., et al. (2021) Minimal tooth preparation for posterior monolithic ceramic crowns: Effect on the mechanical behavior, reliability and translucency [22]. | Enamel | 84.1 | 0.33 |
| | Dentin | 18.6 | 0.32 |
| | Periodontal ligament | 0.069 | 0.45 |
| | High-translucency zirconia: YZHT0 | 210 | 0.33 |
| | Zirconia-reinforced lithium silicate: Vita Suprinity | 65.6 | 0.23 |
| | Hybrid ceramic: Vita Enamic | 34.7 | 0.28 |
| | Resin cement | 7.5 | 0.25 |
| Dal Piva, A.O., et al. (2019) Influence of substrate design for in vitro mechanical testing [23]. | Zirconia-reinforced lithium silicate: Vita Suprinity | 70 | 0.23 |
| | Acrylic resin | 2.7 | 0.35 |
| | Dentin root | 18.6 | 0.32 |
| | Epoxy root | 18 | 0.30 |
| | Resin cement | 6 | 0.30 |

| The Reference | Young's Modulus and Poisson's Ratio are Reported for the Following Materials in the Text | Young's Modulus [GPa] | Poisson's Ratio |
|---|---|---|---|
| Dartora, N.R. et al. (2021) Mechanical behavior of endocrowns fabricated with different CAD/CAM ceramic systems [24]. | Leucite-reinforced vitreous ceramic (IPS Empress CAD) | 65.3 | 0.20 |
| | Lithium disilicate-reinforced vitreous ceramic (IPS and max CAD) | 102.5 | 0.21 |
| | Vitreous ceramic reinforced with lithium silicate and zirconium oxide (VITA Suprinity PC) | 102.9 | 0.19 |
| | Monolithic zirconia (ZirkOM SI) | 206.3 | 0.24 |
| | Bone marrow | 1.37 | 0.30 |
| | Dentin | 18.6 | 0.31 |
| | Periodontal ligament | 0.05 | 0.45 |
| | Gutta-percha | 0.14 | 0.45 |
| Fraga, S., et al. (2018) Does Luting Strategy Affect the Fatigue Behavior of Bonded Y-TZP Ceramic? [26] | Zirconia Y-TZP: Lava Frame | 205 | 0.32 |
| | Epoxy resin: epoxyd-platte | 14.9 | 0.31 |
| | Stainless steel ring/sphere | 190 | 0.27 |
| Guilardi LF, et al. (2021) The influence of roughness on the resistance to impact of different CAD/CAM dental ceramics [27]. | Zirconia-reinforced lithium silicate glass-ceramic ZLS: VITA Suprinity | 70 | - |
| | Lithium disilicate glass-ceramic LD: IPS e.max CAD | 95 | - |
| | Yttria-stabilized tetragonal zirconia polycrystal ceramic YZ:VITA YZ-55/19 T White | 210 | - |
| | Feldspathic glass-ceramic FC: Vitablocs Mark II | 45 | - |
| | Polymer-infiltrated ceramic network PICN: VITA Enamic: 2 M2-HT | 30 | - |
| Lima, J.M.C., et al. (2021) CAD-FEA modeling and fracture resistance of bilayer zirconia crowns manufactured by the rapid layer technology [33]. | Interface resin cement: G10 | 14.9 | 0.31 |
| | Resin cement: Panavia F | 9.2 | 0.28 |
| | Y-TZP In-Ceram YZ | 209.3 | 0.32 |
| | Ceramic: Triluxe Forte | 70.7 | 0.21 |
| Liu, Y., et al. (2018) Bearing, capacity of ceramic crowns before and after cyclic loading: An in vitro study [34]. | Lithium disilicate: IPS e.max CAD | 100.1 | 0.20 |
| | Resin cement: Panavia F | 18.3 | 0.30 |
| | Resin cement: Variolink II | 8.3 | 0.35 |
| | Resin cement: 3M RelyX ARC | 6.4 | 0.27 |
| | Core ceramic: Lava zirconia | 210.0 | 0.30 |
| | Resin composite: 3M Z100 | 16.0 | 0.24 |
| Monteiro, J.B. et al. (2018) Fatigue failure load of two resin-bonded zirconia-reinforced lithium silicate glass-ceramics: Effect of ceramic thickness [38]. | Zirconia-reinforced lithium silicate glass-ceramics: Suprinity | 65.6 | 0.23 |
| | Zirconia-reinforced lithium silicate glass-ceramics: Celtra Duo | 61.0 | 0.30 |
| | Dentin analogue material | 18.0 | 0.30 |
| | Dual cured resin cement: Variolink N | 8.3 | 0.35 |
| | Stainless steel sphere and supporting ring | 195.0 | 0.30 |
| Nakamura, K., et al. (2018) Critical considerations on load-to-failure test for monolithic zirconia molar crowns [39]. | Zirconia: Lava Plus Zirconia | 230.6 | 0.30 |
| | Resin-based cement: Panavia F2.0 | 10.51 | 0.39 |
| | Resin-based composite designed for a dental CAD/CAM (RC): Lava Ultimate | 13.10 | 0.40 |
| | Aluminum-filled castable epoxy resin (EP): EpoxAcast 655 | 8.59 | 0.37 |
| | Polyoxymethylene-copolymer (POM-C): Ertacetal C, Quadrant | 2.45 | 0.39 |

**Table 2.** *Cont.*

| The Reference | Young's Modulus and Poisson's Ratio are Reported for the Following Materials in the Text | Young's Modulus [GPa] | Poisson's Ratio |
|---|---|---|---|
| Penteado, M.M., et al. (2020) Influence of different restorative material and cement on the stress distribution of ceramic veneer in upper central incisor [40]. | Enamel | 84 | 0.30 |
| | Dentine | 18 | 0.23 |
| | Periodontal ligament | 0.069 | 0.45 |
| | Poliurethane fixation cylinder: acquired from the Laboratory of Bioengineering at São Paulo State University (Unesp/São José dos Campos) | 3.6 | 0.30 |
| | Cement agent 1: low elastic modulus resinous cement | 10 | 0.30 |
| | Cement agent 2: medium elastic modulus resinous cement | 18 | 0.30 |
| | Cement agent 3: high elastic modulus resinous cement | 26 | 0.30 |
| | Hybrid ceramic: Vita Enamic | 30 | 0.30 |
| | Zirconia-reinforced lithium silicate: Vita Suprinity | 70 | 0.30 |
| | Lithium disilicate: IPS Emax press | 95 | 0.30 |
| Peskersoy, C et al. (2022) Finite element analysis and nanomechanical properties of composite and ceramic dental onlays [41]. | Enamel | 84.1 | 0.33 |
| | Dentin | 18.6 | 0.31 |
| | Pulp | 0.0028 | 0.45 |
| | Cementum | 4.4 | 0.31 |
| | Periodontal ligament | 0.0689 | 0.45 |
| | Cortical bone | 14.5 | 0.6 |
| | Spongiose bone | 1.37 | 0.30 |
| | Luting cement: RelyX U200 | 7.17 | 0.32 |
| | Conventional composite resin: Tescera ATL 2 | 8.03 | 0.31 |
| | Composite resin block: Cerasmart | 10.36 | 0.30 |
| | Hybrid ceramic block: Vita Enamic | 34.56 | 0.29 |
| | Hybrid ceramic block: Vita Suprinity | 210.1 | 0.29 |
| Ruan, W., et al. (2022) Optimal cuspal coverage of ceramic restorations using CAD/CAM: Biomechanical characteristic analysis by 3D finite element analysis and in vitro investigation [43]. | Enamel | 84.1 | 0.33 |
| | Dentin | 18.6 | 0.32 |
| | Zirconia-reinforced lithium silicateceramic: Vita Suprinity (VS) | 104.9 | 0.21 |
| | Cancellous bone | 1.37 | 0.30 |
| | Cortical bone | 10.7 | 0.30 |
| | Periodontal ligament | 0.0689 | 0.45 |
| | Flowable resin composite: Surefil SDR | 7 | 0.25 |
| | Gutta-percha | 0.00069 | 0.45 |
| Soares, P.M., et al. (2021) Load-bearing capacity under fatigue and FEA analysis of simplified ceramic restorations supported by Peek or zirconia polycrystals as foundation substrate for implant purposes [44]. | Yttria-stabilized tetragonal zirconia polycrystal (YZ) IPS e.max ZirCAD MO | 210 | 0.31 |
| | Polyetheretherketone (Peek) Ceramill PEEK | 4 | 0.4 |
| | Polymer-infiltrated ceramic network (PICN) VITA Enamic; 2 M2-HT | 30 | 0.28 |
| | Zirconia-reinforced lithium silicate glass-ceramic (ZLS): VITA Suprinity, A2 HT PC 14 | 105 | 0.21 |
| | Lithium disilicate glass-ceramic (LD): IPS e.max CAD; LT A2/C14 | 95 | 0.25 |
| | Translucent zirconia (TZ): IPS e.max ZirCAD MT Multi | 200 | 0.31 |
| | Resin cement: Multilink N | 7.5 | 0.30 |
| | Stainless steel ring/sphere | 190 | 0.27 |

**Table 2.** *Cont.*

| The Reference | Young's Modulus and Poisson's Ratio are Reported for the Following Materials in the Text | Young's Modulus [GPa] | Poisson's Ratio |
|---|---|---|---|
| Yang, J., et al. (2022) Comparison of stress distribution between zirconia/alloy endocrown and CAD/CAM multi-piece zirconia post-crown: three- dimensional finite element analysis [48]. | Dentin | 18.6 | 0.31 |
| | Periodontium | 0.05 | 0.45 |
| | Cortical bone | 13.7 | 0.30 |
| | Trabecular bone | 1.37 | 0.30 |
| | Zirconia: e.max ZirCAD | 210 | 0.24 |
| | NiCr alloy multi-piece post-crown: KENNAMETAL | 188 | 0.33 |
| | Glass ionomer cement: Ketac Cem Easymix | 7.56 | 0.35 |
| | Resin cement: Variolink II composite cement | 8.3 | 0.35 |
| Zamzam, H., et al. (2021) Load capacity of occlusal veneers of different restorative CAD/CAM materials under lateral static loading [50]. | Enamel | 84 | 0.30 |
| | Dentin | 18.6 | 0.30 |
| | Stainless steel alloy | 200 | 0.30 |
| | Hybrid ceramic: Vita Enamic | 30 | 0.30 |
| | Lithium disilicate: IPS e.max CAD | 95 | 0.30 |
| | Translucent zirconia: Bruxzir | 210 | 0.30 |
| Zheng, Z., et al. (2021) Biomechanical behavior of endocrown restorations with different CAD-CAM materials: A 3D finite element and in vitro analysis [51]. | Zirconia-reinforced lithium silicate glass ceramic *: Vita Suprinity | 104.9 | 0.21 |
| | Lithium-disilicate glass-ceramic blocks *: IPS e.max CAD | 102.7 | 0.22 |
| | Hybrid ceramic with a dual ceramic-polymer network structure *: Vita Enamic | 37.8 | 0.24 |
| | Resin nano ceramic *: Lava Ultimate | 12.7 | 0.45 |
| | Nano ceramic resin hybrid CAD/CAM blocks *: Grandio blocs | 18.0 | 0.26 |
| | Enamel | 84.1 | 0.33 |
| | Dentin | 18.6 | 0.31 |
| | Spongious bone | 1.37 | 0.30 |
| | Cortical bone | 10.7 | 0.30 |
| | Periodontal ligament | 0.068 | 0.45 |
| | Flowable resin: SDR | 7.0 | 0.25 |
| | Gutta-percha | 0.00069 | 0.45 |
| Zheng, Z., et al. (2022) Influence of margin design and restorative material on the stress distribution of endocrowns: a 3D finite element analysis [52]. | Enamel | 84.1 | 0.33 |
| | Dentin | 18.6 | 0.31 |
| | Spongious bone | 1.37 | 0.30 |
| | Cortical bone | 13.7 | 0.30 |
| | Periodontal ligament | 0.07 | 0.45 |
| | Gutta-percha | 0.00069 | 0.45 |
| | Zirconia-reinforced glass-ceramic: Vita Suprinity | 104.9 | 0.21 |
| | High-leucite content ceramic: IPS Empress | 65.5 | 0.20 |
| | Nano ceramic resin hybrid CAD/CAM blocks *: Grandio blocs | 18.0 | 0.26 |
| | Termoviscous bulk-fill composite: VisCalor bulk | 12.3 | 0.28 |
| | PEEK: Coprapeek Light | 3.7 | 0.40 |
| | Zirconia-Toughened Alumina *: In-Ceram Zirconia | 200 | 0.31 |
| | Flowable resin: SDR | 7.0 | 0.25 |
| | Resin cement: not mentioned * | 7.4 | 0.35 |

* This descriptive sentence is not mentioned in the article's text.

## 4. Discussion

Because of three major advancements, research using finite element analysis is gaining popularity. Firstly, models were typically modeled using volumetric data via abstract image presentations, with dimensions derived from anatomy book images or average tissue form

values. Now, actual tomographic images, scans, or magnetic resonance imaging of the jaws, soft tissue, and prosthetic and restorative elements provide more realistic volumetric data. These data may also include data with varying densities [53,54]. Secondly, although actual data acquisition techniques mandate increased working and computing times, more advanced software and different technical steps are now available to meet these increased requirements [3,4]. Finally, as this review shows, more and more commercially marketed materials have their Young's modulus and Poisson's ratio published along with more detailed soft tissue modeling methods.

One search engine, PubMed, was considered for this narrative review because it accesses many indexed databases of references and abstracts, primarily MEDLINE, and it was sufficient to draw conclusions. The terms used in initial attempted searches was the combined term "finite element analysis". Later, the term "finite element" was used as many studies had replaced the word "analysis" with the following words: method, simulation, modelling, evaluation, study, design-based approach, shape optimization, technique. This might be driven by the need to find word alternatives, which can pass undetected by plagiarism-detecting systems upon publication. Dental zirconia was not specified at this first search step. Thus, zirconia search results were found in articles interested in facial or hip implant materials for orthopedic osseointegration, zygomatic implants combined with obturators, pediatric restorations, occlusal splints, and mouth guards. They were thus excluded, as mentioned above.

Designing an FEA study that has a paired mechanical test using commercial materials requires the following information: Young's modulus and Poisson's ratio. Accessing the studies openly through PubMed may prove critical for researchers who have limited access to subscription-based journals. This was obtainable for less than half of the articles. The abstracts of inaccessible articles serve as a substitute for determining if the article is relevant.

In abstracts, the word "FEA" was commonly used but the commercial name of the material was not. This might be because it is a requirement for some journals not to include commercial names in the abstract. Nevertheless, this may not be considered a limitation for researchers because it was observed that when using any word (e.g., the commercial name) as a search term in PubMed, the search results showed that the search engine will detect that term inside the manuscript text, references, or tables, even if the term was not included in the title or abstract. So, for new researchers, it is sufficient to enter the keywords "finite element" and the commercial name of the product they intend to use, and this will produce search results of actual papers that contain those commercial names within their text. Accessing the full text of the article remains essential because the term may be preset only in the references or introduction.

The term "zirconia" is another critical keyword in this review. Although all included studies used that term, only one-third of them included the chemical composition of their material. It is essential to verify the amount of zirconia and its stabilization form (e.g., yttria-stabilized zirconia: 3, 4, 5, 6, or even 8 mol% $Y_2O_3$). This directly affects the classification of the material, its strength values, and thus the clinical indication for its use. It also affects the translucency of the material [55,56].

Another major issue is the translucency and microstructure of commercial materials, which were reported in less than one-third of the papers. Despite having the same commercial name, many aesthetic commercial ceramic brands have various product lines represented by letters following the initial commercial name. For the various translucencies, letters such as UT (ultra translucency), HT (high translucency), MT (medium translucency), and LT (low translucency) are used. Shade codes can refer to many shade systems, the most prominent among them being A1, A2, A3, etc. from the classical Vita shade guide. The number of layers in the disc or block might range from two (incisal–gingival; 2M) to six, with variations in the shade and translucency. These subtypes vary in translucency by changing mixture percentages, introducing new materials, adding pigments for additional shade gradients, or changing sintering parameters such as heating rate, maximum temper-

ature, and holding and cooling speeds [56]. Differing the translucencies and shades leads to different mechanical strengths [56].

Sintering parameters influence mechanical qualities by influencing crystal growth, density, monoclinic phase, and pore shrinking. Increasing the sintering temperature increased the density and yielded higher translucency. Decreasing the sintering time yielded smaller grain sizes. The higher the sintering time, the larger grain size [57,58]. A combination of a high sintering temperature with a short sintering time increases the flexural strength of zirconia [58]. Altering the grain size eventually lets that material interact with light differently. If the final grain and the light wavelengths are in a similar range, light scattering rises; conversely, when grain size is substantially greater than light wavelength, light scattering decreases regardless of wavelength. The larger grain size zirconia ceramics have reduced translucency [56]. The grain size should be 80 nm or finer to produce a zirconia ceramic with the translucency of dental porcelains [56]. Increased yttrium content at the grain boundary increases translucency. Half of the 5 mol% yttria-stabilized tetragonal polycrystalline (5YTZP) crystals formed cubic phases [59]. These are more isotropic phases with less light scattering at grain borders, more light radiation in all directions, and less light scattering at grain boundaries. These, however, have decreased flexural strength and fracture toughness [60]. Multilayered translucent monolithic zirconia was also successful in mimicking tooth color gradients by layering different translucencies and shades, i.e., including various pigments (such as ferric oxide, erbium, neodymium, etc.). These metallic oxides were considered contaminants because they were found to affect the microstructure of the ceramic and reduce surface hardness [21]. Others found that the pigmentations did not affect the flexural strength or hardness but altered the translucency and contrast ratio property of the ceramic [61–63].

This review compiled the reported values of Young's modulus and Poisson's ratio in Table 2. Though the goal was just for zirconia products, the researcher deemed it worthwhile to add other materials mentioned in the same publication to provide the values entirely, having taken advantage of the fact that they were already referenced and cited.

The primary goal of this review was to check that the Young's modulus and Poisson's ratio are correctly referenced, as opposed to using the generic numbers that have been reported for zirconia material for a long time (i.e., 210 GPa, 0.3, respectively) [64]. Although these values were referenced in almost two-thirds of the articles, there were certain difficulties that needed to be addressed. Some lists featured references for only a few of the materials; therefore, not all of them had a reference. In some cases, the reference for the commercial product was the manufacturer's sources or the ISO standards sheet. The researcher followed the first direct reference for those values that had their reference as a published study. Several of these investigations reported the previously mentioned "old" general zirconia values. Several referenced even older references, but they were not tracked back in this research since it was outside the scope of the review. Finally, although these Young's modulus and Poisson's ratio values had a reference, they were not consistent for the same material. For example, Vita Suprinity had the following values: 65, 65.5, 65.6, 70, 104.9, 105, 210.9 GPa and a ratio of 0.21, 0.23, 0.29, 0.30 [17,22–24,27,38,40,41,51,52]. This also applies to many living tissue values such as spongy bone and material such as gutta-percha. Reasons for this variance should be further explored, and researchers are strongly recommended to search PubMed for relevant FEA studies using the commercial name of the product they intend to use. They should then trace the references of Young's modulus and Poisson's ratio values for the material of interest to ensure that the values reported are valid.

It is worth finally mentioning and appreciating the few references that have reported these values, obtained through in-house tests and designed specifically for new commercial dental zirconia CAD/CAM materials [15,16,65].

## 5. Conclusions

The purpose of this narrative review was met: to compile and reference all available research presenting Young's modulus and Poisson's ratio data for commercial CAD/CAM zirconia restorative materials. This list of data would directly benefit researchers interested in this field of research and would also serve as a platform from which new research may proceed into materials not yet included. New studies are invited not only to examine the ever expanding list of new commercial materials, but also to test their product lines featuring various translucencies, shades, and numbers of layers.

**Funding:** This research received no external funding.

**Institutional Review Board Statement:** Not applicable.

**Informed Consent Statement:** Not applicable.

**Data Availability Statement:** Research articles reviewed here are present in PubMed–indexed journals.

**Conflicts of Interest:** The author declares no conflict of interest.

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
