# Peer review of "A Narrative Review of Recent Finite Element Studies Reporting References for Elastic Properties of Zirconia Dental Ceramics"

_ceramics, doi:10.3390/ceramics6020052_

Round 1

Reviewer 1 Report

Ceramics, MDPI

ceramics-2237901

Title: A Literature Review of Recent Finite Element Studies reporting the elastic properties of commercial zirconia dental ceramics.

Summary: Zirconia is nowadays preferable as fixed partial denture (FPD) material. New generations with different properties and design options are of great interest. In vitro studies based on mechanical testing and studies including FEA are published but still the knowledge and understanding of different material testing and analysis are lacking.

Abstract:

“The aim is to search recent FEA studies that report these values for newly developed commercial CAD CAM zirconia prosthetic\restorative materials.”

Based on the background given, the aim can be rephrased and be interpreted as objective. Why is the search needed? What´s the outcome, more than elucidate the data?

Is it not to apply the findings (correlate material testing with FEA)? The aim corresponds to the conclusion, and the conclusion is in my opinion nothing that adds something new (these conclusions are already stated/already known).

“Methods: A PUBMED search was performed for published English articles between 2018 and 2023, using the keywords (FEA, finite element, zirconia). Studies included were reporting commercial materials’ Young's modulus and Poisson's ratios. (3) Results: Following the removal of duplicates, /…/”

The search included just one database, PubMed. Why did you make this limitation, by choosing a subject specific database? It is clarified, but not motivated in the discussion.

This study is conducted as a review, but based on the aim, material and methods, results it can be considered as a narrative review.

By choosing PubMed, follows the choice of using keywords and MeSH-terms. Based on the info given in the Abstract the following is addressed: Why did you do the limitation of using keywords, and the specific three ones? And finally, how is it possible to have duplicates when you search in one database? Have you done the search for each keyword, or with a search block (combining the boolean operators OR, AND or NOT)?

(This is more clear in the Material and Methods, consider to rephrase)

Introduction:

There are some parts in the manuscript that need clarification. Some examples follow:

“Restorations or prostheses materials, such as implants /…/”

The restorations or prostheses materials - does this not include that the restorations are made by prostheses materials? The sentence contains an aim error.

“An anisotropic material has material properties that differ depending on the direction. The materials are typically modeled as homogeneous, isotropic, and linearly elastic.”

Suggestion, add However/Still/some kind of follow up to the statement of properties and be clear. (An anisotropic material has material properties that differ depending on the direction. Still, the materials are typically modeled for FEA as homogeneous, isotropic, and linearly elastic.)

 “Others suggest pairing confirmative studies with the FEA. These can be; actual mechanical tests, conventional clinical model analysis, preclinical tests, and long-term clinical studies - the gold standard”

Consider rephrasing and be clear about what can be paired and what is gold standard. Are the confirmatory studies what is mentioned in the following sentence? Are all gold standard?

“So, if the generic numbers for the Young's modulus and Poisson's ratios are used in the FEA calculations, some might argue that drawing conclusions, or comparisons between the FEA results and the confirmation study results - using commercial materials - is unrealistic.”

Is it not a fact, there cannot be a comparison between different study designs? For instance, FEA and clinical studies are not comparable, as there cannot be a comparison between in vitro studies and clinical studies without considering limitations. Be clearer on what can be paired and what is gold standard.

Aim:

“The aim of this review is to search for recent FEA studies that report the Young’s modulus and Poisson’s ratios for actual, and newly developed commercial CAD CAM zirconia prosthetic or restoration materials. This allows us to conclude which studies could be recommended as references, when designing FEA studies that are paired with mechanical confirmation tests using CAD CAM zirconia materials.”

There is an objective and aim in this part, however the aim is still vague in my opinion.

Material and Methods

The search block is more specific, than presented in the abstract, still the search is missing information based on reproduction. Was the search and inclusion done by one person? What kind of further limitations was used? What was the inclusion criteria? Was there a bias analysis performed?

Results

The total amount of included articles; 37 after exclusion, but 27 referenced. This is not clear, including the figure.

 “3.2. Essential information extracted from the texts of the selected articles.

The following information were extracted from the texts of the selected articles (Table 1):”

This part in text and in table format is not needed. Consider to only use the table.

“Table 2 contains a list of the articles that report references for the Young's modulus and Poisson's ratios of newly developed commercial CAD CAM zirconia prosthetic or restoration materials, and other materials also listed in the Young's modulus and Poisson's ratios’ tables.”

Other materials included - How is this addressed to the aim/objectives?

Discussion:

“It is essential to verify the amount of zirconia, its stabilization form (3,4,5,or 6 Y-TZP). This directly affects the classification of the material, thus the clinical indication for its use [55].”

The classification in this sentence and following Discussion section, is wrong, the zirconia can be stabilized with more than 3 mol % and the tetragonal phase will not be dominant, meaning that the Y-TZP is not correctly used with 4-,5-, 6-.

Conclusions: (see previous comments)

The conclusion can be re-written with a clearer message to the readers, researchers, clinicians.

Reviewer 2 Report

Dear Author

Thank you for your efforts. Kindly find the comments on your work in the attached PDF. 

Kind regards

Reviewer 3 Report

This manuscript by Layla A. Abu-Naba'a, titled “A Literature Review of Recent Finite Element Studies reporting the elastic properties of commercial zirconia dental ceramics” presented an important review for the mechanical properties of zirconia ceramics for their dental applications. The review topic targeted by the author are of high market demand and is helpful to the community. However, the manuscript lacks proper organization and a few important points have been mentioned below:

Title: Some of the words are capitalized while the others are not. Needs to be fixed.

Abstract: Sections such as "Background", " Methods" and others are not necessary in the abstract. An abstract should be a continuous flow of text summarizing the current research gap and the solutions presented in the manuscript. 

Page 1, line 33, "In-vitro studies that simulate isolated or combined...clinical studies": Citation is missing in this fact mentioned. Some other sentences in the manuscript also have the same issue of missing citation. For example, Page 2, Line 55, "Non-living mechanical structures...on the research questions.". The author needs to cite every fact that has been mentioned in the manuscript with proper references. Similar cases in the rest of the manuscript needs to be checked and fixed.

Page 3, line 103, "...papers should have been published in English no earlier than 2018.": Grammar on this sentence needs to be checked. The section of the sentence should be something in the lines of "...papers published in English and later than 2018 were selected.". Similar minor grammatical errors in the rest of the manuscript needs to be checked and fixed. 

Page 4, line 150: Review methodology needs to be a part of the materials and methods section. Results should talk about the findings that were obtained from the manuscripts that are reviewed here.

Page 9, line 214, "Using the search terms...": This is a repeated information that has already been mentioned in the methods section. The next two paragraphs of this section also has a lot of methodology related descriptions mentioned. Discussion section should solely discuss the results that are collected from the manuscripts reviewed. These issues are making this draft lack correct organization that needs to be fixed. Editing the content and making them a part of the correct sections is very important.

Page 10, line 263, "Sintering parameters influence mechanical qualities...": This and the  rest of the phenomenon presented needs to be discussed in detail. How are the parameters effecting and what is the author's thought on this? The readers will need detailed information to be benefitted from this review work. The same needs to be done with the rest of the phenomenon mentioned in this paragraph.

Round 2

Reviewer 1 Report

Ceramics, MDPI

ceramics-2237901 revision

Title: A Literature Review Of Recent Finite Element Studies Reporting 2 The Elastic Properties Of Commercial Zirconia Dental Ceramics

Comments: The author have done an extensive revision; however language revision and further clarifications are needed.

 Example of language revision

Abstract:

“The aim of this narrative review was find recent FEA studies that report these values for newly developed commercial.”

 Introduction:

“To ensure the use of newly developed materials and treatment modalities of materials, that have not been thoroughly tested, simulations and non-invasive studies are used, thus further reduction of the time and cost requirements

 “One well-known example is dental implant research, where finite element analysis (FEA) studies aided in the investigation of stresses exerted on the the peri-implant region and in the components of implant-supported restorations.”

 “Stress may generate strain if it is strong enough to surpass the strength of the sample being stressed thus, strain is characterized by an alteration in form or dimensions caused by applied forces (deformation).”

Comment: What does form, or dimensions mean? The form change when the dimensions change, is it not the same?

 “Poisson's ratio is described as the ratio of a substance's change in width per unit width to its length alteration per unit length as a consequence of strain, whereas Young's modulus /…/”

 “This leads to the most common drawback of FEA, from the clinical perspective, is that many features that directly affect model accuracy are neglected or ignored by multiple simplifications and assumptions.”

 Aim:

The aim is revised and clarified.

 Material and Methods

The same applies for the M&M, however, remove the section ( 143-154) that explains the choice of database, PubMed, it belongs to the discussion.

 Results

The aim is revised and clarified

 Discussion:

“Even though they have the same commercial name, many aesthetic commercial products have several translucencies, shades and layering denoted by letters after the commercial name. For the translucencies they use letters as (LT, MT, HT ...etc.), shades as (A2,A3, B2, C2 ..etc.) /.../”

The classification of translucency is in translucency of different grades and not defined as several translucencies, this need to be clarified.

“Increasing the sintering temperature, increased density and higher translucency. Decreasing the sintering time yielded smaller grain sizes. Increasing the sintering time, the larger grain size.”

Why is this part added? And why as the first sentence in the section that describes one of many properties – interesting choice. Furthermore, following part refereeing to ref. 56. This is just one study, reading newly published systematic reviews, they report that there is a difference between different types of zirconia, brands and how they have been processed. Important to state that this is only based on one study and why you have this limitation sintering – translucency in combination of FEA.

 “Finally, although referenced, values of Young's modulus and Poisson's ratio were not consistent for the same material. For example Vita /…/”

There is something that is missing in this sentence.

Conclusions:

The conclusion has been revised, still there is a mixture of statements and summary of results. What is the actual conclusions of this study?

Reviewer 2 Report

Dear Author

Thank you for your submitting your revised version of the manuscript. Another PDF was attached for your reference. The main issue with the review is that it did a quite good job finding and collecting the article and failed to analyse them. The discussion needs to be deeper, connects the numbers and facts in the papers you ended up with but I could not find answers where I should have found.

Thank you and regards 

Round 3

Reviewer 2 Report

Thank you for your efforts. Please find the attached PDF with comment.

Regards
